# Does Nitrofurantoin Improve the Portfolio of Vets against Resistant Bacteria in Companion Animals?

**DOI:** 10.3390/antibiotics12050911

**Published:** 2023-05-15

**Authors:** Cristina Vercelli, Michela Amadori, Graziana Gambino, Giovanni Re

**Affiliations:** Department of Veterinary Sciences of Turin, University of Turin, 10124 Torino, Italy; graziana.gambino@unito.it (G.G.); giovanni.re@unito.it (G.R.)

**Keywords:** nitrofurantoin, veterinary medicine, dog, cat, companion animals, pharmacokinetic, pharmacodynamic

## Abstract

In clinical practice in dogs and cats, antimicrobials are frequently used, sometimes overused or misused, increasing antimicrobial resistance (AMR). In order to limit the phenomenon, laws have been enacted and guidelines for prudent and rational use of antibiotics have been developed. Interestingly, old molecules such as nitrofurantoin could be used to achieve therapeutic success and overcome AMR. To better understand the suitability of this molecule in veterinary medicine, the authors performed a revision of the literature, searching on PubMed and entering the following keywords: nitrofurantoin, veterinary medicine, dog, and cat connected by the Boolean operator “and”, without restrictions on the date of publication. Thirty papers were finally selected. It is possible to appreciate that papers dealing with nitrofurantoin have been written from the early 1960s to the middle of the 1970s, and then a long period passed without publications. Only at the beginning of the new century, nitrofurantoin was included or was sometimes the focus of papers dealing with its efficacy in veterinary medicine, mainly in the treatment of urinary tract infections. One recent paper dealt with pharmacokinetic features, and none was dedicated to pharmacokinetic/pharmacodynamic integration or modeling. Nitrofurantoin appears to be still effective against several pathogens that rarely develop resistance to this molecule.

## 1. Introduction

Antimicrobials are widely used to treat bacterial infections in dogs and cats. It has been previously reported that antimicrobials are prescribed to treat skin infections (52%), urogenital infections (11%), respiratory diseases (10%), gastrointestinal diseases (10%), dental diseases (7%), and other types of infections (10%) in dogs [1]. In cats, the situation is slightly different considering the different percentages: antimicrobials are prescribed to treat skin diseases (42%), respiratory diseases (24%), urinary tract infections (16%), periodontal diseases (14%), as perioperative administration (1%), and for other pathologies (3%) [1]. Several concerns about the use, misuse, and abuse of antimicrobials have been reported in recent decades, mainly related to the possibility of a rise in the worldwide phenomenon of antimicrobial resistance (AMR) among animals and the possibility of sharing mechanisms and genes of resistance among animals and humans [2]. Thus, the interest of the entire scientific community has been focused on understanding the mechanisms of resistance, finding new strategies to implement antimicrobial stewardship programs (ASP), and evaluating the pharmacokinetic (PK) and the pharmacodynamic (PD) features of antimicrobial drugs labeled in veterinary medicine in order to better target the therapy [3,4]. As previously mentioned, these aspects may represent a risk for people’s health considering the possibility of AMR sharing between pets and owners and as specific consequences for animals considering that AMR could lead to longer hospitalization of the patient, increased demand for diagnostic tests, and higher healthcare costs for owners [5]. As the release of new antimicrobial molecules in the next few years is not feasible both for human and veterinary medicine, it is important to prioritize the therapeutic interventions of available drugs [6]. This aspect has been specifically considered by the European legislation that restricted the use of antimicrobials in veterinary medicine, confirming the ban on the use of these drugs as growth-promoting agents, and restricting their use in prophylaxis and metaphylactic protocols [7]. More specifically, Article 37, paragraph 5 of European Regulation 6/2019 and the Commission Implementing Regulation (EU) 2022/125 [8], designate antimicrobials or groups of antimicrobials reserved for the treatment of certain infections in humans, with the final aim of better preserving their efficacy for human medicine and to supporting the fight against AMR.

According to the last categorization of antimicrobials performed by the AntiMicrobial Expert Group (AMEG), released in December 2019, all classes have been ranked, allowing a greater distinction in ranking between substances: this made it possible to prevent too many antibiotics from being placed in the highest categories [9]. This categorization ranks the compounds in four categories: Avoid (A), Restrict (B), Caution (C), and Prudence (D) [10]. The first one encompasses drugs that are not authorized as veterinary medicines in the EU, should not be used in food-producing animals, and may be administered to companion animals under exceptional circumstances. Category B includes all antibiotics considered critically important in human medicine, the use of which in animals should be restricted in order to reduce consequences for public health. These compounds should be used only in case of treatment failure with drugs enrolled in categories C and D, and only after the performance of antimicrobial susceptibility tests. Category C includes antibiotics that are not critically important for human health and should be used only in case of treatment failure of antimicrobials encompassed in category D, which represents the first line of intervention. The term “first line” does not mean that the compounds present in this category can be used without control and in any situation: the principle of prudent and rational use must always be the driving force of the prescription of all antimicrobials, including those present here.

Among the compounds present in category D, it is possible to appreciate nitrofuran derivatives, such as furaltadone and furazolidone [10]. Nitrofurantoin belongs to this antibiotic class, and it is considered a urinary tract antiseptic, primarily in small animals, and occasionally in horses, to treat lower urinary tract infections caused by susceptible bacteria such as *Escherichia coli* (*E. coli*), *Klebsiella* spp., *Enterobacter* spp., *Staphylococcus aureus* (*S. aureus*) and *epidermidis* (*S. epidermidis*), *Citrobacter* spp., *Salmonella* spp., *Shigella* spp., and *Corynebacterium* spp. [11]. Nitrofurantoin’s mechanism of action is bacteriostatic but might be bactericidal depending on the concentration of the drug and specifically related to the bacteria causing the infection. The molecular mechanism responsible for the final effect has not been fully elucidated but seems to be related to the inhibition of bacterial enzyme systems, including acetyl coenzyme A [11]. Nitrofurantoin efficacy is increased in an acidic environment, but it is less efficacious against *Proteus* spp., *Serratia* spp., and *Acinetobacter* spp., and totally not efficacious against *Pseudomonas* spp. The PK features of nitrofurantoin demonstrated rapid absorption from the gastrointestinal tract, with increased absorption in fed animals. The therapeutic levels in the systemic circulation are not maintained due to the rapid elimination of the drug from the body through glomerular and tubular filtration: the peak level in urine occurs 30 min after the administration, and approximately 50% of the drug is eliminated in unchanged form [11]. A negligible part of the drug is biotransformed in the liver.

The aim of the present review is to summarize the currently available information in the literature about nitrofurantoin specifically and exclusively related to dogs and cats. The focus is to rediscover an old drug, commonly not prescribed by veterinarians, that might represent a strategic tool for achieving therapeutic success in pathologies having bacterial etiology affecting companion animals.

## 2. Results

At the end of the selection process, 30 papers were considered adequate and have been included in the present review (Figure 1). The resulting papers are presented and discussed in chronological order.

In order to provide readers with better use of the information, tables have been structured presenting the selected papers divided according to whether the topics concerned pharmacokinetic or pharmacodynamic studies (Table 1), according to the bacteria (Table 2), and the organs and systems considered, respectively (Table 3).

## 3. Discussion

According to the papers that have been selected for the present review, it was possible to delineate a timeline, from the first studies dealing with nitrofurantoin till nowadays. Initially, the interest in this molecule was high, but a gap of almost 30 years intervened and nitrofurantoin was considered again for therapeutic purposes only at the beginning of the new century. The interest in nitrofurans, and particularly in nitrofurantoin, began in the early 1960s and it was not only related to investigating the ability of nitrofurantoin to act as a urinary antiseptic but also to be a therapeutic agent in other organs. Buzard and Conklin in 1965 [12] reported for the first time their results regarding the penetration of nitrofurans in cerebrospinal fluid and aqueous humor in dogs. They demonstrated that these drugs, including nitrofurantoin, can be accumulated in these fluids without underlying differences among the different compounds of this class of antimicrobial agents. After some years, Conklin and Wagner investigated the biliary excretion of nitrofurantoin in dogs presenting an experimentally induced hepatic failure: only 20% of the administered amount of the drug was subsequently measured in bile [13]. Dr. Conklin published several papers with different co-authors dealing with the PK of nitrofurantoin in dogs: in some cases, only the titles of these papers are listed in PubMed but no abstract or full-text is available, and thus it was not possible to include them all in the present review. Nevertheless, it is possible to appreciate the interest in investigating the PK features of this molecule over a long time. In the paper by Conklin and Wagner [13], it is possible to find some results of other investigations and it seems clear that, even if the analytical methods had lower sensitivity, specificity, and accuracy compared to the modern techniques, they were able to understand that nitrofurantoin is excreted in the urine in an unmodified form, after oral and intravenous administration. Even if this point was a clear milestone, Dr. Conklin and coworkers also investigated the bile excretion of nitrofurantoin in dogs a few years later [14], due to the fact that in their previous work they observed an hydrocholeresis effect, suggesting an enterohepatic reabsorption. The results of this study did not elucidate this point clearly and the authors were not able to determine the exact amount of intestinal drug re-absorption. In parallel, Wick and colleagues investigated the antibacterial property of a derivative of nitrofurantoin, named 1-(2-Hydroxyethyl)-3-nitro-4-pyrazole carboxamide [16]. These authors performed minimal inhibitory concentration (MIC) determinations and microbiological assays using *Bacillus subtilis* (*B. subtilis*) and then performed acute toxicity assays in mice, rats, and dogs. They obtained an encouraging result considering the high antibacterial efficacy of this compound and the low acute toxicity but, to the authors’ knowledge, this molecule is currently commercially available for research purposes only and it is not registered for clinical use.

After more than two decades of a decreased interest in nitrofurantoin, in 2004, Sannes et al. described for the first time the identification of resistant *E. coli* strains in human and canine species [17]. These authors investigated the sensitivity of *E. coli* isolated from urine of women affected by cystitis or pyelonephritis, and from fecal samples of dogs and healthy human volunteers. The results of test diffusion assays demonstrated resistance against ampicillin, sulfisoxazole, trimethoprim, and trimethoprim-sulfamethoxazole, mainly in samples collected from women compared to those collected from dogs: the reason behind these results may be related to the fact that these molecules were often prescribed by physicians. Moreover, no resistances have been highlighted for nitrofurantoin and other molecules, such as ceftazidime, ciprofloxacin, piperacillin, and tazobactam. Interestingly, these authors concluded that dogs were unlikely to represent a risk for humans since they do not act as reservoirs, but they can acquire resistant *E. coli* from humans [17].

The resistant pattern of *E. coli* was also investigated a few years later during a study performed in the United States using samples collected in companion, food-producing, and wild animals, human septage, and water [18]. The samples were treated in order to determine the presence of bacteria and to isolate and identify *E. coli*. Then, sensitivity to gentamicin, neomycin, streptomycin, chloramphenicol, ofloxacin, trimethoprim-sulfamethoxazole, ampicillin, nalidixic acid, nitrofurantoin, cephalothin, and sulfisoxazole was tested using the disk diffusion method. At that time and in the experimental conditions reported by the authors, the highest percentage of resistance was recorded in swine and poultry fecal samples: almost 60% of the isolated *E. coli* demonstrated resistance against two or more antimicrobial agents, and almost the entire amount of the bacteria isolated from the samples demonstrated resistance against tetracycline [18]. The results concerning nitrofurantoin were encouraging since resistance was reported only for dairy and beef cattle, poultry, and small ruminants, while the *E. coli* isolated from swine, equids, companion animals, farmed deer, wild geese, and human septa still demonstrated sensibility to the drug [18]. The resistant pattern of bacteria colonizing the bowel was the focus of another paper that dealt with the isolation of enterococci from dogs and cats, and the evaluation of their susceptibility against 14 different antimicrobial compounds representative of all antimicrobial classes, including nitrofurantoin [19]. The different resistance features of the isolated bacteria have been compared to those of strains of *Enterococcus faecalis* (*E. faecalis*), *S. aureus*, and *E. coli* that have been used as quality controls. The authors reported that the predominant species identified were *E. faecalis* in dogs and *Enterococcus hirae* (*E. hirae*) in cats, demonstrating significant differences in resistance patterns. In fact, *E. faecalis* isolated in dogs demonstrated resistance against ciprofloxacin, gentamicin, and chloramphenicol, while the isolates from cats demonstrated a high percentage of resistance against nitrofurantoin [19]. This was the first paper reporting this kind of information among all papers considered so far in the present review, but unfortunately, authors did not hypothesize a reason behind this phenomenon, nor did they provide readers with some clinical information about the enrolled animals that could help readers understand the causes of this specific resistance.

In 2010, Pomba and colleagues described the clinical management of a cat affected by a urinary tract infection caused by *Staphylococcus pseudintermedius* (*S. pseudintermedius*) and *E. faecalis* [20]. The cat demonstrated signs and symptoms of recurrent urinary infections and urethral obstruction. After a complete diagnostic procedure and based on the laboratory report, it was clear that both bacteria presented a multidrug resistance pattern and the only sensitivities left were toward nitrofurantoin and a few compounds intended for humans (i.e., teicoplanin and vancomycin) since they represent the last therapeutic resource for serious infection in human medicine. The authors explained that at that time, nitrofurantoin was not authorized in animals, but they chose this compound since it did not belong to the critically important antimicrobials [20]. It was decided to prescribe nitrofurantoin at 4 mg/kg, three times a day (the dose reported in Plumb’s Veterinary Drug Handbook even in the latest edition), for 60 days; signs of clinical improvement were appreciated after 5 days, and further urine samples collected on days 5, 20, 25, and 60 after commencing nitrofurantoin treatment, were culture negative [20]. The paper by Maaland and Guardabassi [21] supported the final consideration proposed by Pomba and coworkers. In fact, they evaluated the MICs of nitrofurantoin in 269 canine and feline isolates of *E. coli* and *S. pseudintermedius*, all resistant or multidrug-resistant strains. The MICs were below the drug concentration reported in canine and feline urine after oral administration of nitrofurantoin. Moreover, these authors evaluated the mutant prevention concentration (MPC), to understand if the perpetuation of the prescription and the administration of nitrofurantoin in human beings could have exerted a selective pressure capable of allowing the emergence of resistant strains: the results highlighted that these factors did not represent an issue for this drug [21]. The authors concluded that the use of nitrofurantoin to treat urinary tract infections in dogs and cats seemed to be promising considering the very low rate of acquired resistance, but they did not encourage its use as a first-line agent considering the poor PK properties and the possibility of inducing hepatic toxicity [21].

Only in 2011, a study confirmed the resistance against nitrofurantoin displayed by *E. faecalis* and *E. faecium* isolated in seven dogs treated for 2–9 days with antimicrobials in a veterinary intensive care unit [22]. The samples were first cultured and identified, and then susceptibility tests were performed: the highest resistances for *E. faecium* were recognized against enrofloxacin, ampicillin, tetracycline, doxycycline, and erythromycin (ranging from 50 to 90%), while less important but still present against gentamicin, streptomycin (around 45%), and nitrofurantoin (26.5%) [22]. *E. faecalis* demonstrated significant resistance against tetracycline, erythromycin, doxycycline, and enrofloxacin, but with minor intensity. No resistance was detected to vancomycin, tigecycline, linezolid, and quinupristin/dalfopristin in either species [22]. Multilocus variable number tandem repeat analysis (MLVA) and pulsed-field gel electrophoresis (PFGE) revealed that six sequence types (STs) originating from five dogs were identical or closely related to STs of human clinical isolates and isolates from hospital outbreaks: according to the results, the authors recommend avoiding close physical contact between pets released from an intensive care unit and their owners in order to limit the spread of AMR [22].

To support the necessity of educating people about hygiene and safe contact with animals in order to limit the spread of AMR, Rubin and Chirino-Trejo [23] investigated the similarities in MICs for 33 different antimicrobials used in human and veterinary medicines on 99 *S. aureus* isolates collected from people and 27 from dogs. The only drug that resulted in exerting an effect was nitrofurantoin, while all other antimicrobials were deemed inadequate. The most worrying results were represented by the fact that inducible clindamycin resistance was found among 78% and 4% of canine and human methicillin-resistant *S. aureus* (MRSA) respectively, and 17% and 25% of canine colonizing and human methicillin-susceptible *S. aureus* (MSSA), respectively [23]. Moreover, the authors were able to conclude that the transmission of *S. aureus* might be bidimensional between people and dogs, and thus, also in this case, considerations about client education were proposed [23].

In 2015, a paper by Dorsch and colleagues [24] evaluated antimicrobial susceptibility in feline bacterial urinary tract infections using a retrospective design over a period of 10 years in Germany. The most common pathogen identified was *E. coli*, followed by Streptococcus and Staphylococcus species, Enterococcus and Micrococcaceae: the majority of the strains were susceptible to nitrofurantoin, and also to amoxicillin and clavulanic acid, enrofloxacin, and gentamicin. According to the results obtained, the authors concluded that the use of nitrofurantoin increased significantly over time but did not affect the efficacy of the drug [24]. In the same year, a paper by Lund and coworkers dealt with the antimicrobial susceptibility in bacteria isolates from cats presenting urinary tract infections in Norway [25]. *Escherichia coli*, Staphylococcus species, Enterococcus species, and Streptococcus species were the most frequently detected. The highest susceptibilities were recorded for enrofloxacin (92%), trimethoprim/sulfonamide (91%), and nitrofurantoin (89%) followed by tetracycline, ampicillin, amoxicillin and clavulanic acid, spiramycin, fusidic acid, and lincomycin with progressively decreasing percentages [25]. The high susceptibility to enrofloxacin is surprising compared to the other papers considered so far. Specifically considering nitrofurantoin, no differences have been highlighted with other references present in the literature. The authors were focused on this topic because they wanted to understand if some specific resistance pattern was occurring in Norwegian cats considering the high prevalence of urinary tract infections, but they did not find any significant differences with the data presented in the literature or any other clinical reason [25].

The necessity to understand if bacterial strains and mechanisms of resistance can be shared between humans and animals was the driving force of a paper by Humphries and colleagues [26]. These authors evaluated the MICs of 115 *S. pseudintermedius* isolated in both species and it was possible to assess that 33% was methicillin-resistant (mainly related to the presence of mecA gene), and a significant percentage was resistant to doxycycline, clindamycin, and trimethoprim-sulfamethoxazole. All isolates were susceptible to nitrofurantoin and a few other antimicrobials labeled only for humans [26].

Similar results have been obtained by Priyantha and colleagues, who evaluated the antimicrobial susceptibility of *S. pseudintermedius* in healthy dogs in Canada from 2008 to 2015 [27]. They collected rectal and pharyngeal swabs to isolate the bacteria of interest and then performed a polymerase chain reaction (PCR) to sequence the mecA gene. Only 7 isolates out of 221 were identified as methicillin-resistant and no resistance was identified against fluoroquinolones, nitrofurantoin, tigecycline, vancomycin, quinupristin–dalfopristin, linezolid, or daptomycin; however, also in this case, the majority of sensitivity was described for drugs licensed only for humans (i.e., tigecycline, vancomycin, quinupristin–dalfopristin, linezolid, or daptomycin). In fact, only nitrofurantoin is used both in human and veterinary medicine, and it is not listed as a critically important antimicrobial for humans. The authors concluded that, according to the comparison between the data obtained at the beginning of the study and those at the end, the frequency of resistance to all antimicrobials increased globally [27].

Proceeding with the chronological exposition, it is possible to appreciate the growing awareness of antibiotic resistance as a result of technological and scientific progress, which has made it possible to understand the different mechanisms underlying the acquisition and spread of resistance. Considering the articles cited and considered up to now, it was possible to identify only sporadic references to sophisticated and advanced methods for determining antibiotic sensitivity. Considering instead the articles of the last six years, there is an appreciably drastic change. Shimizu and colleagues analyzed 90 extended-spectrum β-lactamase (ESBL)-producing isolates of extraintestinal pathogenic *E. coli* (ExPEC) from companion animals using PCR and DNA sequencing after the execution of susceptibility test [28]. Even in this case, the old but gold nitrofurantoin demonstrated a very high rate of efficacy because 96.7% of ESBL isolates were still susceptible to this drug, while only 10% were susceptible to enrofloxacin and 63.3% to amoxicillin and clavulanic acid. The genetic analysis showed that 92.2% of isolates were positive for CTX-M-type genes [28]. The authors concluded that nitrofurantoin and a few other drugs might be used to treat companion animals infected with ExPEC-producing CTX-M-type ESBLs.

The paper by Teichmann-Knorrn and colleagues [29] updated the data obtained by Dorsch et al. a few years before [24] about the antimicrobial resistance pattern of cats affected by urinary tract infections. The authors designed a retrospective study to review clinical reports of cats in the period between July 2009 and November 2014, evaluating the susceptibility profile, calculating the impact factor (i.e., a parameter that describes the likelihood that a bacterial uropathogen would be sensitive to an antimicrobial drug based on in vitro susceptibility testing), and comparing the data with those obtained five years before. According to the isolation, the most representative bacterial populations were *E. coli*, Staphylococcus species, Enterococcus species, Streptococcus species, and *Proteus mirabilis* (50%, 22.9%, 15.1%, 3.6%, and 2.6%, respectively), demonstrating susceptibility against imipenem, nitrofurantoin, gentamicin, and amoxicillin–clavulanic acid [29]. The authors discussed their results and established that the antimicrobial resistance patterns were not significantly different compared to the previous retrospective study. Despite this, the authors underlined the importance of limiting the empirical treatment of feline urinary tract infections, and introduce for the first time (considering the papers enrolled in the present review) the concept of location: bacteria prevalence and susceptibility can significantly vary among regions and countries, even among hospitals, and thus recommendation and guidelines should be as circumscribed as possible [29]. From this paper on, it is quite common to appreciate the location where the study takes place in the titles of the papers. This is the case of the paper by Abbas et al. [30], who immediately declared to readers that the aim of the investigation was to evaluate the role of pets as reservoirs of ESBL-producing *E. coli* in Pakistan. More than one hundred fecal samples collected from dogs, cats, their owners, and veterinary professionals from veterinary clinics were used to perform the isolation of ESBL *E. coli* and to conduct further analysis with PCR to identify the presence of *bla*CTX-M genes and CTX-M groups I and II [30]. The study interestingly put into correlation the percentage of *E. coli* isolated in dogs, cats, and their respective owners and veterinarians: the percentage of identification of *E. coli* was extremely high in all groups (more than 80%). A total of 17.4% of the phenotypically resistant *E. coli* exhibited a multidrug resistance profile, resulting in resistance to ampicillin, cefotaxime, ciprofloxacin, and nitrofurantoin. *bla*CTX-M and *bla*CTX-M-1 were identified in the same bacteria and retained responsibility for the multidrug-resistant pattern [30]. Thus, in this specific study, almost one *E. coli* out of five acted as multidrug ESBL, not responsive either to β-lactams or to nitrofurantoin. The results exposed in the present paper are in line with those concerning the resistance to nitrofurantoin displayed by other Gram-negative bacteria: at that time authors were not able to justify the results obtained [22].

By contrast, the paper written by Suay-Garcıa and colleagues seemed to reestablish the effectiveness of nitrofurantoin against ESBL *E. coli* [31]. The authors collected 325 fecal samples randomly from different species of healthy animals frequenting humans: dogs, cats, monkeys, horses, sheep, goats, falcons, and pigeons. Only 34 isolates contained *E. coli* and all were recognized to be ESBL [31]. The susceptibility test demonstrated worrying results about amoxicillin, aztreonam, cephalosporins, nalidixic acid, ciprofloxacin, and trimethoprim/sulfamethoxazole that showed a very high percentage of resistance (100% for β-lactams and more than 70% for the others). Nitrofurantoin was shown to be effective against all isolates, together with ertapenem, minocycline, imipenem, meropenem, amikacin, fosfomycin, and colistin [31]. The authors declared their concerns about the possibility of transmission of ESBL *E. coli* between animals and humans: the same consideration was also formulated by Abbas et al. [30].

Technological and scientific progress has made it possible to apply molecular biology and gene sequencing techniques aimed at evaluating and identifying the genes responsible for the acquisition of antimicrobial resistance. Considering specifically the resistance against nitrofurantoin, Li et al. decided to evaluate in their review the specific role of the OqxAb efflux pump, believed to be responsible for the phenomenon of multidrug resistance in different bacterial species [32]. They reviewed 87 papers downloaded from PubMed and ISI Web of Science and they concluded that the *oqxAB* gene is mainly located in chromosomes and/or plasmids flanked by IS26-like elements: it is largely expressed in clinical isolates of Enterobacteriaceae and *Klebsiella pneumoniae* and it is responsible for the resistance against quinoxalines, quinolones tigecycline, and nitrofurantoin. Moreover, the *oqxAB* gene can be co-spread with other antimicrobial resistance genes, such as *bla*CTX-M, rmtB, and aac(6′)-Ib [32]. The reporting of the co-spread of oqxAB and *bla*CTX-M genes could be a reasonable link with the previously cited paper [30], which held the *bla*CTX-M gene responsible for *E. coli* (thus a Gram-negative bacteria) exhibiting a multidrug resistance profile against ampicillin, cefotaxime, ciprofloxacin, and nitrofurantoin. Li and co-workers deeply analyzed the specific topic of the genetic mechanisms of resistance against nitrofurantoin: they reported that the resistance to nitrofurantoin in *E. coli* is primarily due to mutation in the nitroreductase genes (called *nfsA* and *nfsB*) but also related to the expression of *oqxA* and *oqxB*. These last two genes have been recognized as being responsible for a decreased susceptibility or a complete resistance to nitrofurantoin both in humans and animals clinically affected by urinary tract infections [32].

The original article by Yu and colleagues [33] reported the results obtained in a prospectively designed study, focused on investigating the susceptibility of bacteria responsible for urinary tract infections in companion animals referred to the veterinary teaching hospital of Beijing (China) from 2016 to 2018. They established that *E. coli* and *Klebsiella* spp. were the most represented Gram-negative bacteria, while *S. aureus* was the most represented among Gram-positive bacteria. Multidrug resistance was detected for 39% of *E. coli* and 51.5% of *Staphylococcus* spp. isolates, while the highest resistances were identified against beta-lactams and erythromycin, for *E. coli* and *Staphylococcus* spp., respectively. Only 6% of *Staphylococcus aureus* demonstrated resistance against nitrofurantoin [33]. The authors cited some genetic mechanisms of resistance in the discussion, but they were mainly focused on the *mcr-1* gene, without presenting a specific correlation with nitrofurantoin resistance: they limited their discussion to the increasing rate of resistance reported for bacteria responsible for urinary tract infection in their region [33].

In the panorama of Gram-negative bacteria and nitrofurantoin, the paper by Zhen et al. represents a unique case of its kind [34]. In this original article, the authors reported the results obtained in the evaluation of the sensibility of *Bartonella henselae* (*B. henselae*) against 14 different drugs (i.e., amikacin, azithromycin, cefuroxime, ciprofloxacin, clofazimine, daptomycin, disulfiram, doxycycline, gentamicin, methylene blue, miconazole, nitrofurantoin, rifampin, and trimethoprim/sulfamethoxazole) and 25 possible combinations. The authors performed susceptibility tests and they evaluated the disruption of biofilm induced by single drugs or their combinations. Ciprofloxacin, gentamicin, and nitrofurantoin were the most active in inhibiting the proliferation of *B. henselae* while the combinations of azithromycin/ciprofloxacin, azithromycin/methylene blue, rifampin/ciprofloxacin, and rifampin/methylene blue completely eradicated the biofilm of *B. henselae* after 6 days of treatment [34]. Nitrofurantoin was one of the most active agents against the stationary phase of these bacteria. This report is interesting since it considered nitrofurantoin in the treatment of *B. henselae*, an intracellular Gram-negative bacteria responsible for serious and worldwide zoonosis, transmitted mainly by direct contacts such as animal scratches (in particular of cats) and bites, or by some arthropods such as sand flies, lice, fleas, biting flies, and ticks [34]. Unfortunately, the authors did not discuss the PK features of nitrofurantoin, and thus they do not provide readers with any hypothesis about how therapy may be planned and applied in vivo, locally (at the point where the scratch was received, for example), or systemically. In the conclusions, the authors wrote only that this paper laid the foundations for future studies and evaluations for a clinical application, of which however no further traces have been found in recent literature, considering the search criteria used to write the present review.

A 2020 paper by Leuin specifically aimed to evaluate in a retrospective study the efficacy of nitrofurantoin in 14 dogs presenting recurrent urinary tract infections and referred to the University of Wisconsin Veterinary Care from July 2013 to January 2019 [35]. The results were in line with those previously described in the literature and demonstrated that 12 dogs out of 14 had a successful outcome, including bacteriologic and clinical cures. Therapeutic failure occurred only in two dogs due to the targeted uropathogen developing progressive nitrofurantoin resistance [35]. The dogs received an oral median dose of 4.3 mg/kg every 8 h ranging from 7 to 28 days, in accordance with the current dosage regimens presented in Plumb’s Veterinary Drug Handbook [11]. The authors proposed a very interesting and well-written discussion about the use of nitrofurantoin in dogs and they underlined that little is known about the reasons related to the limited clinical use of this drug, and that the majority of gastrointestinal side effects are anecdotal and not supported by detailed scientific reports. The PK data of nitrofurantoin considered in the present paper are the same as reported in the papers of the 1970s, while the discussion of PD features is easier considering that several in vitro or in vivo investigations have been performed over the decades to understand the susceptibility of different bacteria. Despite the amount of data available, no breakpoint interpretations for nitrofurantoin in dogs existed either in 2020 or nowadays, and clinical interpretations of susceptibility tests in dogs are still uncertain and are based on breakpoints used in people, which may not be appropriate for canine urinary isolates [36].

Trying to gain additional information on the more well-known use of nitrofurantoin in urinary tract infections, Dégi et al. investigated the prevalence of drug-resistant *Salmonella* spp. in 85 fecal samples derived from client-owned cats originating from Timisoara (Romania) [37]. *Salmonella* spp. was detected in 16 samples that were molecularly tested for the presence of the *invA* gene. Three serotypes have been identified: *Salmonella* (*Sal.*) *enteritidis*, *Sal. typhimurium*, and *Sal. kentucky*. All of the tested strains showed strong resistance towards cefazolin, cefepime, ceftazidime, and ceftriaxone. Eight Salmonella strains out of 16 demonstrated resistance to nitrofurantoin. The authors did not comment further on the results regarding nitrofurantoin, and they wrote that their data should be considered as a warning for public health, considering the potential risk of having AMR spread from cats to vulnerable people [37]. It must be remarked that in this specific case, the samples taken into consideration are few compared to other investigations and that it is hard to generalize.

In the same year and in the same location, a paper by Janos and colleagues described the multidrug-resistant pattern of Staphylococci isolated in dogs housed in the shelter in Timisoara (Romania), including healthy subjects or dogs affected by dermatological pathologies [38]. The authors collected 78 samples and performed bacteria isolation, confirming the identification of 43 Staphylococci using different methods. Subsequently, they evaluated the susceptibility profile against a panel of antimicrobial drugs: the majority of strains (37 out of 43) were resistant to benzylpenicillin, kanamycin, and tetracycline, chloramphenicol (29/43, 67.44%), while 11 strains were resistant to nitrofurantoin [38]. According to the results obtained in the study, the authors defined that 8 strains must be considered methicillin-resistant Staphylococci (MRS), according to their susceptibility profile and the identification of the *mecA* gene, and this can pose a serious and concrete risk for public health considering that MRS has to be considered as a zoonotic agent [38]. They explained that shelters are small places with high promiscuity of individuals, thus facilitating the transmission of MRS. Consequently, the operators who work inside the kennels, the health personnel, and the possible adopters, should be carefully educated in order to avoid colonization by these extremely resistant bacterial strains and their spread [38]. Specifically related to the clinical application of their results, they did not provide suggestions or comments, and similarly to Dégi et al. [37], it is hard to draw generalized conclusions considering the small number of samples included in both studies.

A paper by Palomino-Farfàn and co-workers [39] focused on investigating *S. schleiferi* subspecies coagulans in Peru, because it is frequently isolated in canine otitis externa and pyoderma, and even in cases of zoonoses. The authors collected 331 swabs from dogs with otitis externa and pyoderma that were cultured on agar for bacterial isolation, and subsequently biochemical and molecular identification were performed. A polymerase chain reaction was conducted in order to evaluate the presence and prevalence of the *mecA* gene [39]. *Staphylococcus schleiferi* was identified in 34 and 12 samples in otitis externa and pyoderma, respectively. Almost 50% of the bacteria were resistant to fluoroquinolones, but this data did not surprise the authors considering that this class of antimicrobial is commonly used and prescribed in the case of otitis externa and pyoderma in Peru [39]. The 40% of *S. schleiferi* isolated from otitis externa were resistant to methicillin, and 85.29% presented the *mecA* gene, while only one isolate from pyoderma expressed the same gene [39]. In both pathological conditions, the most encouraging results have been shown with nitrofurantoin: also in this paper, in the discussion or in other parts of the manuscript, considerations about dosage regimen, possible routes of administration, PK/PD characteristics, and hypotheses related to the clinical outcome are missing.

A massive investigation was reported in 2022 by Aurich and colleagues of the prevalence and antimicrobial resistance of canine and feline bacterial uropathogens in Germany [40]. They analyzed 1862 urine samples and isolated 962 uropathogens that underwent susceptibility tests against 15 different antimicrobial agents, matrix-assisted laser desorption/ionization-time of flight (MALDI-TOF) identification, and several experimental procedures. The manuscript is relevant not only considering the specific aim of the present review but also for the quantity and quality of the data and explanations provided, as regards the methods performed (with extremely precise references to internationally validated guidelines), the results obtained, and the clinical and therapeutic considerations, but also in light of recent European legislation [7]. The authors explained that nitrofurantoin is recommended as a second-line antimicrobial in sporadic bacterial cystitis, and should only be considered in cases of urinary tract infection caused by drug-resistant bacteria, due to its toxicity and poor pharmacokinetics [40]. The authors compared their results with those in the literature and confirmed that also in their specific experimental condition, the efficacy of nitrofurantoin against multidrug-resistant isolates was excellent, but they did not recommend nitrofurantoin to treat *Klebsiella* spp. and *E. faecium* due to the high resistance recorded in their study. It should also not be applied in infections caused by *Proteus* spp. and *Pseudomonas* spp., as these bacteria are intrinsically resistant to nitrofurantoin [40]. Consideration of the safety profile is provided: the authors explained that differently from the original formulation, the macrocrystalline formulation of nitrofurantoin actually commercially available rarely causes gastrointestinal side effects, and that veterinarians must seriously take into consideration owner compliance since the drugs must be administered every 8 h due to its low plasma concentration [40]. According to the most recent European regulation [7], nitrofurantoin has to be used as an off-label drug under the Cascade principle, as there is no equivalent product licensed for veterinary usage in Europe [40].

A deep insight into the clinical application of nitrofurantoin is provided by Sposato and colleagues, who described how a successful therapeutic outcome was obtained by treating multidrug-resistant *E. faecium* [41]. This bacterium was isolated in a 10-year-old Shetland sheepdog affected by cholecystitis, referred for aspecific signs and symptoms. After cholecystectomy and the collection of samples and biopsies, it was possible to isolate two strains of multiresistant *E. faecium*, susceptible to nitrofurantoin. The authors decided to administer nitrofurantoin following the standard dosage regimen of 5 mg/kg, every 8 h, orally, achieving a complete recovery and normalization of blood parameters after 77 days [41]. The authors explained that the multidrug resistance of these two strains is commonly due to the exchange of mobile genetic elements among Enterococci, and that the presence of the same genotypes of *Enterococcus* spp. isolated from various animals and humans has been previously reported, suggesting interspecies transmission of strains [41].

In 2022, Ekstrand et al. [15] published a paper entirely dedicated to the PK features of nitrofurantoin in dogs. Nitrofurantoin was orally administered to eight healthy beagles every 8 h for five days, at a standard dosage regimen (4.4–5.0 mg/kg). Then, plasma and urine samples were collected repeatedly. The nitrofurantoin plasma and urine concentrations were measured using ultra-high-precision liquid chromatography coupled to tandem mass-spectrometry (UPCL-MS). The data were analyzed using a non-compartmental pharmacokinetic model [15]. In plasma, the median C*max* was 2.1 μg/mL, T*max* was 2 h, the terminal rate constant was 0.9 per h, and the terminal half-life was 0.8 h. In urine, the median C*max* was 56 μg/mL, T*max* was 1 h, and the terminal half-life was 4.3 h [15]. The authors commented on the fact that no adverse effects were observed in the clinical evaluations of the dogs, and no hematological or biochemistry alterations were appreciated during the entire experiment, thus suggesting a high safety profile in the clinical application of this drug [15]. This is in contrast to a previous work by Maaland and Guardabassi, which underlined the high frequency of adverse reactions [21], but it is supported by Leuin et al. [35].

The paper by Ekstrand and co-workers represents a novelty among the papers that have been considered in the present review: after almost 60 years, some authors decided to perform a PK study of nitrofurantoin after Conklin and colleagues [13,14].

Last but not least, a recent paper by Tumpa and colleagues [42] reported the susceptibility profile of 12 *E. faecium* and 17 *E. faecalis*, isolated from urinary samples of 22 dogs and 7 cats in Croatia. *Enterococcus faecium* isolates were significantly more resistant to penicillin, ampicillin, nitrofurantoin, and ciprofloxacin when compared to *E. faecalis* [42]. This paper is worthy of attention since the authors circumscribed and localized the data about the efficacy of nitrofurantoin in Croatia, without generalizing, confirming that susceptibility to this drug and the prevalence of resistance might be similar or different to that reported in other parts of the world [17,23,30], and thus confirming that bacterial behavior and susceptibility can be strongly influenced by climatic conditions, prescribing habits, or legislative requirements.

## 4. Materials and Methods

A systematic review has been performed considering only the archive of PubMed. The authors decided to enter the keywords nitrofurantoin, veterinary medicine, dog, and cat connected by the Boolean operator “and”. No restriction on the date of publication was applied. The resulting papers have been checked to find duplicates, and further selected if they were written in English (other languages were considered as an exclusion criterion) and the abstract was available. The last part of the selection process was performed by reading the abstracts: only those dealing with dogs and cats were considered. The entire workflow with the different passages of the selection process is presented in Figure 1.

## 5. Conclusions

Considering the literature that has been summarized in this review, it is possible to conclude that:-Only three papers provided sound explanations about the resistance against nitrofurantoin in dogs and cats affected by susceptible bacteria;-The major clinical application and the most frequent reason for the prescription of nitrofurantoin was to treat urinary tract infections in dogs and cats;-Other clinical applications have been rarely reported (for example, the treatment of Bartonella or Salmonella);-Some papers did not discuss the possible clinical application of nitrofurantoin and simply add this drug to complete the list of susceptibility tests;-It is encouraging that the global trend of AMR did not affect nitrofurantoin efficacy from the beginning of the century till nowadays;-Only one paper deals with the PK features of nitrofurantoin but considers only healthy dogs, without clinical or pathological issues;-Localization is important to understand differences in the prevalence of resistant bacteria;-Despite nitrofurantoin being one of the oldest clinically available compounds, it is still one of the most efficacious drugs for the treatment of urinary tract infections;-No breakpoints have been established in dogs and cats and no in silico simulation or PK/PD modeling have been reported till nowadays.

## Figures and Tables

**Figure 1 antibiotics-12-00911-f001:**
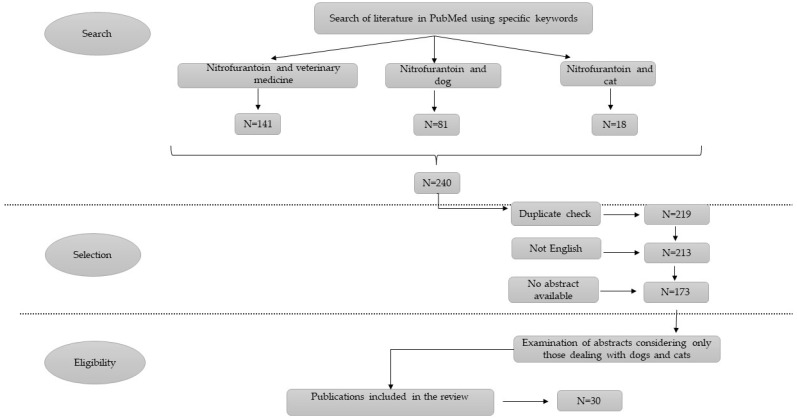
The scheme represents the workflow followed by the authors to select papers included in the present review. The criteria for eligibility are also presented.

**Table 1 antibiotics-12-00911-t001:** The table presents the papers considered in the present review divided between pharmacokinetic (PK) and pharmacodynamic (PD) studies.

Type of Study	References
PK	[12,13,14,15]
PD	[16,17,18,19,20,21,22,23,24,25,26,27,28,29,30,31,32,33,34,35,36,37,38,39,40,41,42]

**Table 2 antibiotics-12-00911-t002:** The table presents the papers considered in the present review divided by the Gram staining (Gram-positive or Gram-negative) and subdivided according to the bacterial species.

Gram Staining	Bacterial Species	Reference/s
Gram-positive	*Bacillus subtilis*	[16]
	*Staphylococcus aureus*	[23,33]
	*Staphylococcus pseudintermedius*	[20,21,26,27]
	*Staphylococcus* spp.	[24,25,29,38]
	*Staphylococcus schleiferi*	[39]
	*Streptococcus* spp.	[24,25,29]
Gram-negative	*Bartonella henselae*	[34]
	*Enterobacteriaceae*	[32]
	*Enterococcus faecalis*	[19,20,22,42]
	*Enterococcus faecium*	[22,40,41,42]
	*Enterococcus hirae*	[19]
	ESBL	[28,30,31]
	*Escherichia coli*	[17,18,21,24,25,29,33]
	*Klebsiella pneumoniae*	[32,33]
	*Klebsiella* spp.	[40]
	*Proteus mirabilis*	[29]
	*Proteus* spp.	[40]
	*Pseudomonas* spp.	[40]
	*Salmonella enteritidis*	[37]
	*Salmonella kentucky*	[37]
	*Salmonella* spp.	[37]
	*Salmonella typhimurium*	[37]

**Table 3 antibiotics-12-00911-t003:** The table presents the papers selected for the present review divided by organs and systems considered.

Organs/System	References
Digestive system	[19,30,31,41]
Genitourinary system	[17,20,21,24,25,29,35,37,40,42]
Integumentary system	[38,39]
Respiratory system	[27]
Whole body	[18,22,23,24]

## Data Availability

All data are presented along the text.

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
