# Peer review of "Does Nitrofurantoin Improve the Portfolio of Vets against Resistant Bacteria in Companion Animals?"

_antibiotics, 2023, doi:10.3390/antibiotics12050911_

Round 1

Reviewer 1 Report

Antimicrobial resistance (AMR) is one of the most important human- and animal health-related issues worldwide. Reviewing the potential use of under-utilized drugs, such as nitrofurantoin, in veterinary medicine will provide timely information to overcome global AMR problems. This review selected the related papers and gave comments for each papers. However, in this current form, the manuscript is nothing more than a simple collection of summaries of related paper. Reorganization of the manuscript by related subject (resistant bacteria, isolation source, rate of resistance, resistant mechanism, etc.) and summarize the analysis as a table are highly recommended.

Minor editing of English language required.

Author Response

Dear R1, thank you for the time that you spent reviewing our manuscript. Your comments were precious for us: we took inspiration from your suggestions to structure 3 tables to make the content presented in the review more usable for readers. We divided the selected papers according to whether the topics concerned pharmacokinetic or pharmacodynamic studies (Table 1), according to the bacteria considered (Table 2), and to the organs and systems considered (Table 3) and we presented the related references.

We left the manuscript organized as a timeline, describing from the oldest to the newest papers found in literature about nitrofurantoin use in companion animals. This organization for us is the best to appreciate the ups and downs of scientific interest in this molecule. In fact, nitrofurantoin was the subject of numerous studies between the 60s and 70s and then remained in oblivion for almost 30 years. Only at the time of the emergence of antibiotic resistance, nitrofurantoin returned to being of clinical interest for pets and in scientific research.

Reviewer 2 Report

This manuscript deals with an important topic in medicine - the control of resistance of bacterial populations to antibiotics - specifically, it deals with the possibility of treating bacterial infections in dogs and cats with nitrofuration. Nitrofurantion was previously used in veterinary medicine - primarily to treat urinary infections in small animals. Currently it is used less and is often replaced by other antibiotics. However, some of these have limited use (categories A - C) according to the new classification of antimicrobials in animals. Nitrofurantion and its derivatives belong to category D, which, when used prudently for treatment, poses the least risk in terms of the development of resistance. I find the topic discussed very interesting and important and the results discussed may be a good guide for veterinarians in clinical practice.

The manuscript is based on a literature search of publications in the PubMed database and the results of the publications found are discussed in detail. The manuscript is appropriately structured and the section titles correspond to the content.

I recommend that the article be published in Antibiotics without necessary modifications.

Author Response

Dear R2, thank you for your comments! I sincerely thank you, on behalf of all authors, for the time that you spent to read our manuscript.

We have to modify something according to the comments and suggestions of the other two reviewers but we are delighted that you appreciated our work.

Regards, Dr. C. Vercelli 

Reviewer 3 Report

The current systematic review sheds light on using nitrofuran on vets. The current review should be much improved.

Elaborate on the aim of this study clearly in the introduction

What nitrofurantoin is used for? clinical uses in animals and its impact on human 

Please provide a table (s) to summarize the results of the presented studies (30 studies)

Please provide more information about the mode of action of nitrofuran and what is the impact of such a study on the public health

Could you provide a figure to deliver your findings easier,  Figure 1: font is very small

Write bacteria in italic

The conclusion should be written as a paragraph 

The English can be improved

Author Response

Dear R3, thank you for the time that you spent reviewing our manuscript.

In order to answer every single comment, we provide hereby a point-to-point response.

The current systematic review sheds light on using nitrofuran on vets. The current review should be much improved.

Thank you for your comment. We did our best to improve the quality of the manuscript according to your comments and those provided by R1. R2 evaluated our manuscript suitable for publication without further modifications.

Elaborate on the aim of this study clearly in the introduction

Thank you for your comment. We improved the explanation of the aim of the study and we slightly modified the text in order to better circumscribe the context.

What nitrofurantoin is used for? clinical uses in animals and its impact on human

Thank you for your comment that permit us to clarify some points. First of all, we modified the title in order to relate the topic only to companion animals. Then, the importance on public health and human beings is explained along the text considering that the drug is classified in category D of AMEG classification, since it is a molecule with a low impact on selective pressure on antimicrobial resistance. Moreover, several explanations are provided along the discussion considering that several authors considered different mechanisms of acquisition of resistance but it seems that the resistance against nitrofurantoin is extremely specific and does not cause the resistance against other antibiotic classes. This last point is particularly important also for human beings: if bacteria would develop resistance or acquire resistance against nitrofurantoin, even if resistant bacteria can spread among pets and humans, this will not cause the acquisition of resistances against other antimicrobials.

Completely different is the panorama about the use of nitrofurantoin in food producing animals: it is forbidden by the present European Regulation (EU Reg.37/2010, Table 2 Prohibited substances). This argument is out of the topic of the present review and will be considered in another manuscript in the few months.

Please provide a table (s) to summarize the results of the presented studies (30 studies)

Thank you for your comment. We structured 3 tables, summarizing the selected papers considered for the present review. This should answer to your comment and to a comment of R1.

Please provide more information about the mode of action of nitrofuran and what is the impact of such a study on the public health

Thank you for your comment. The pharmacodynamic features of nitrofurans/nitrofurantoin is provided along with the introduction considering the most important text books about Veterinary Pharmacology and Therapeutics. The answer about implications in public health is related to what we answered above.

Could you provide a figure to deliver your findings easier,  Figure 1: font is very small

Thank you for your suggestion: we increased the font and we substituted Fig.1 in the manuscript. We will address your comment to the Editorial Office in order to find a graphical solution in accordance with Journal style.

Write bacteria in italic 

Done. We will address your comment and our correction to the Editorial Office in order to be in accordance with Journal style.

The conclusion should be written as a paragraph 

Conclusions are already presented as a paragraph (i.e. number 5) and placed after the Material and Method section, according to journal style requirements.

According to R1 and R3 suggestions, the English style was edited by a native speaker.

Round 2

Reviewer 1 Report

Complete reorganization of the manuscript by related subject (resistant bacteria, isolation source, rate of resistance, resistant mechanism, etc.) is required. The manuscript is just a simple collection of summary of papers. It is very hard for readers of this review to get useful information. In current form, the impact for the field would be very limited.

No issues in quality of English.

Author Response

Dear R1,

we would like to receive a more polite response since we use polite and gentle sentences. Nevertheless, we are sorry that you are the only one among three reviews that did not understand that the manuscript is organized in a chronological manner.

The other two reviewers got it and both approved the manuscript in the present form.

Moreover, you just copied and pasted some sentences of your previous comments without giving us feedback about the tables that we drawn taking inspiration from your comments. According to the new version of the manuscript, tables would be useful to readers to reach quickly the most important topics treated along the text.

We had seriously considered all your comments and we provided a careful and detailed response, clearly explaining the aim of the manuscript and giving the reason why it is functional to maintain this organization.

We will wait the opinion of the Academic Editor and of the Editorial Office.

Regards.

Reviewer 3 Report

The authors addressed the raised points.

The language is understood and accepted

Author Response

Thank you! We are delighted that you appreciated the improvement of the manuscript!

Regards!